# Energy and Entropy Production of Nanofluid within an Annulus Partly Saturated by a Porous Region

**DOI:** 10.3390/e23101237

**Published:** 2021-09-22

**Authors:** Zehba A. S. Raizah, Ammar I. Alsabery, Abdelraheem M. Aly, Ishak Hashim

**Affiliations:** 1Department of Mathematics, College of Science, King Khalid University, Abha 62529, Saudi Arabia; ababdallah@kku.edu.sa; 2Refrigeration Air-Conditioning Technical Engineering Department, College of Technical Engineering, The Islamic University, Najaf 54001, Iraq; ammar_e_2011@yahoo.com; 3School of Mathematical Sciences, Faculty of Science & Technology, University Kebangsaan Malaysia, Bangi 43600, Malaysia; ishak_h@ukm.edu.my; 4Department of Mathematics, Faculty of Science, South Valley University, Qena 83523, Egypt

**Keywords:** natural convection, horizontal annulus, nanofluid-porous cavity, Darcy-Forchheimer model, entropy production

## Abstract

The flow and heat transfer fields from a nanofluid within a horizontal annulus partly saturated with a porous region are examined by the Galerkin weighted residual finite element technique scheme. The inner and the outer circular boundaries have hot and cold temperatures, respectively. Impacts of the wide ranges of the Darcy number, porosity, dimensionless length of the porous layer, and nanoparticle volume fractions on the streamlines, isotherms, and isentropic distributions are investigated. The primary outcomes revealed that the stream function value is powered by increasing the Darcy parameter and porosity and reduced by growing the porous region’s area. The Bejan number and the average temperature are reduced by the increase in Da, porosity ε, and nanoparticles volume fractions ϕ. The heat transfer through the nanofluid-porous layer was determined to be the best toward high rates of Darcy number, porosity, and volume fraction of nanofluid. Further, the local velocity and local temperature in the interface surface between nanofluid-porous layers obtain high values at the smallest area from the porous region (D=0.4), and in contrast, the local heat transfer takes the lower value.

## 1. Introduction

Natural or free convection has vast, important applications in the industry field, such as electronic systems cooling [1], energy collectors solar [2], heat exchangers [3], electric ovens [4], desalination solar systems [5], melting [6,7] and freezing processes [8,9], etc. The heat transfer rate in natural convection is low as a result of the fluid velocity connected with natural convection is minimum. In natural convection, the fluid velocity is relatively low due to the fluid movement caused by the buoyancy force. Therefore, the heat transfer becomes weak also. For these reasons, a technique must be found to enhance the natural convection of heat transfer inside enclosures. One of the methods is to add nanoparticles to the base fluids, or use the porous media inside the enclosures.

The porous medium, for example, is a hard sedimentary rock, wood, bread, and a lung or a substance consisting of a solid structure with an interconnected vacuum. The fluid flow through the interconnected void depends on the porosity and the permeability of the media. Regarding shape and size, when the distribution of pores is unequal, the media is called natural porous media, for example, sandstone and bones [10]. Over the years, numerous researchers have made great progress in the subject of porous media inside enclosures. Porous media are perfect for improving conduction inside enclosures. On the other hand, it causes resistance to the fluid flow and weakens convection. Nanomaterials of infinitesimal sizes play a successful role in the heat transfer, so the nanofluids are used simultaneously with porous media to improve heat transfer [11,12,13]. Toosi and Siavashi [11] investigated the natural convection inside a square cavity partially filled with porous media and of Cu–water nanofluid. They used Corcione’s model for a two-phase model and Darcy–Brinkman–Forchheimer to simulate fluid flow in porous media. They found that the convection becomes weak by using nanofluid and porous media, and also the conduction can be strengthened. Baghsaz et al. performed a study on natural convection heat transfer and entropy generation inside a porous cavity filled with Al${}_{2}$O${}_{3}$/water nanofluid. They found that the natural convection heat transfer, circulation, and irreversibility of the production increased clearly by increasing the Rayleigh number (Ra) and Darcy number (Da). Miroshnichenko et al. [14] simulated the natural convection in an open cavity containing porous layers and filled with alumina/water nanofluid, which penetrated the cavity from the open boundary. They used a single-phase nanofluid approach and the Brinkman-extended Darcy model for porous layers. Their results showed that the heat transfer rate was increased by the expansion of the range between the wall and the porous layer. Cho [15] analyzed the heat transfer of natural convection inside a porous cavity filled with nanofluid and containing a partially-heated vertical wall. He showed that for high values of Da and Ra, convection heat transfer dominates. Thus, the entropy generation and mean Nusselt number increases, whereas the Bejan number decreases. Using the finite volume method, Selimefendigil and Öztop [16] performed a study on magnetohydrodynamics forced convection of carbon nanotube of water nanofluid in a U-shaped cavity containing a porous region under the impact of a wall rippling. They illustrated that the variation of the Reynolds number and permeability of the porous medium affected the flow field and heat transfer. Using a Local Thermal Non-Equilibrium (LTNE) approach, Mehryan et al. [17] reported the fluid flow and heat transfer of natural convection inside a porous enclosure filled with Ag-MgO/water nanofluids. Alsabery et al. [18], Raizah et al. [19], Aly and Raizah [20], Javaherdeh et al. [21], Rao and Barman [22], Wang et al. [23] and Esfe et al. [24] executed numerical imitation studies to analyze the natural convection heat transfer of a nanofluid in porous cavities. They illustrated that one could enhance the heat transfer performance by assuming both porous media and nanofluid.

As noticed from the earlier research surveys, and to the best of the authors’ knowledge, there is no outcome with the convection flow within horizontal annulus filled with nanofluid superposed porous layers. Accordingly, this work introduces the understanding of nanofluid superposed horizontal annulus porous layers via the fluid flow and heat transfer distributions.

## 2. Mathematical Formulation

The steady natural convection and heat transfer of water-Al${}_{2}$O${}_{3}$ nanofluid within an annulus by radius rr and having an inner hot ($T_{h}$) cylinder by radius *r* is explained in Figure 1. The analyzed composite annulus cavity is divided into two segments, the first one (outer portion) is filled by nanofluid and the second one (inner portion) is loaded by a porous region saturated by a nanofluid. The outer surface is fixed with a cold temperature of $T_{c}$. The edges of the domain (except for the interface surface between the nanofluid-porous layers) are supposed to remain impermeable. The mixed liquid inside the composite cavity performs as a water-based nanofluid holding Al${}_{2}$O${}_{3}$ nanoparticles. The Forchheimer–Brinkman-extended Darcy approach and the Boussinesq approximation remain appropriate. In contrast, the nanofluid phase’s convection and the solid matrix are in the local thermodynamic equilibrium condition. The set of porous media applied in the following output is glass balls ($k_{m}=1.05$ W/m·${}^{\circ }$C). Examining earlier specified hypotheses, the continuity, momentum and energy equations concerning the Newtonian laminar flow survive, formulated as:

For the nanofluid region:(1)$$\begin{array}{ccc} & & \frac{\partial u_{nf}}{\partial x}+\frac{\partial v_{nf}}{\partial y}=0,\;\;\;\;\;\;\;\;\;\;\;\;\;\;\;\;\;\;\;\;\; \end{array}$$(2)$$\begin{array}{ccc} & & u_{nf}\frac{\partial u_{nf}}{\partial x}+v_{nf}\frac{\partial u_{nf}}{\partial y}=-\frac{1}{\rho_{nf}}\frac{\partial p}{\partial x}+\frac{\mu_{nf}}{\rho_{nf}}\left(\frac{\partial^{2}u_{nf}}{\partial x^{2}}+\frac{\partial^{2}u_{nf}}{\partial y^{2}}\right),\;\;\;\;\; \end{array}$$(3)$$\begin{array}{ccc} & & u_{nf}\frac{\partial v_{nf}}{\partial x}+v_{nf}\frac{\partial v_{nf}}{\partial y}=-\frac{1}{\rho_{nf}}\frac{\partial p}{\partial y}+\frac{\mu_{nf}}{\rho_{nf}}\left(\frac{\partial^{2}v_{nf}}{\partial x^{2}}+\frac{\partial^{2}v_{nf}}{\partial y^{2}}\right)+\beta_{nf}g\left(T_{h}-T_{c}\right), \end{array}$$(4)$$\begin{array}{ccc} & & u_{nf}\frac{\partial T_{nf}}{\partial x}+v_{nf}\frac{\partial T_{nf}}{\partial y}=\frac{k_{nf}}{{\left(\rho C_{p}\right)}_{nf}}\left(\frac{\partial^{2}T_{nf}}{\partial x^{2}}+\frac{\partial^{2}T_{nf}}{\partial y^{2}}\right).\;\;\;\;\;\;\;\; \end{array}$$For the porous region: (5)$$\begin{array}{ccc} & & \frac{\partial u_{m}}{\partial x}+\frac{\partial v_{m}}{\partial y}=0,\\ & & \frac{\rho_{nf}}{\varepsilon^{2}}\left(u_{m}\frac{\partial u_{m}}{\partial x}+v_{m}\frac{\partial u_{m}}{\partial y}\right)=-\frac{\partial p}{\partial x}+\frac{\mu_{nf}}{\varepsilon }\;\left(\frac{\partial^{2}u_{m}}{\partial x^{2}}+\frac{\partial^{2}u_{m}}{\partial y^{2}}\right) \end{array}$$(6)$$\begin{array}{ccc} & & \;\;-\left(\frac{\mu_{nf}}{K}u_{m}-\frac{1.75}{\sqrt{150}\varepsilon^{3/2}}\frac{\rho_{nf}u_{m}\left|u\right|}{\sqrt{K}}\right),\\ & & \frac{\rho_{nf}}{\varepsilon^{2}}\left(u_{m}\frac{\partial v_{m}}{\partial x}+v_{m}\frac{\partial v_{m}}{\partial y}\right)=-\frac{\partial p}{\partial y}+\frac{\mu_{nf}}{\varepsilon }\;\left(\frac{\partial^{2}v_{m}}{\partial x^{2}}+\frac{\partial^{2}v_{m}}{\partial y^{2}}\right) \end{array}$$(7)$$\begin{array}{ccc} & & \;\;-\left(\frac{\mu_{nf}}{K}v_{m}-\frac{1.75}{\sqrt{150}\varepsilon^{3/2}}\frac{\rho_{nf}v_{m}\left|u\right|}{\sqrt{K}}\right)+{\left(\rho \beta \right)}_{nf}g\left(T_{h}-T_{c}\right), \end{array}$$(8)$$\begin{array}{ccc} & & u_{m}\frac{\partial T_{m}}{\partial x}+v_{m}\frac{\partial T_{m}}{\partial y}=\frac{\varepsilon k_{nf}}{{\left(\rho C_{p}\right)}_{nf}}\left(\frac{\partial^{2}T_{m}}{\partial x^{2}}+\frac{\partial^{2}T_{m}}{\partial y^{2}}\right).\;\;\;\;\;\;\;\;\;\; \end{array}$$The subscripts nf and *m* view the nanofluid layer and porous layer. *x* and *y* are the fluid velocity elements, $\left|u\right|=\sqrt{u^{2}+v^{2}}$ denotes the Darcy velocity, g displays the acceleration due to gravity, ε signifies the porosity of the medium and *K* is the permeability of the porous medium, which is determined as [25]:(9)$$K=\frac{\varepsilon^{3}d_{m}^{2}}{150{\left(1-\varepsilon \right)}^{2}}.$$

Here, $d_{m}$ represents the average particle size of the porous region.

The thermo-physical characteristics regarding the adopted nanofluid for 33 nm particle-size will be prepared as [26]: (10)$$\begin{array}{ccc} & & {\left(\rho C_{p}\right)}_{nf}=\left(1-\phi \right){\left(\rho C_{p}\right)}_{f}+\phi {\left(\rho C_{p}\right)}_{p},\;\;\;\;\;\;\;\; \end{array}$$(11)$$\begin{array}{ccc} & & \rho_{nf}=\left(1-\phi \right)\rho_{f}+\phi \rho_{p},\;\;\;\;\;\;\;\;\;\;\;\; \end{array}$$(12)$$\begin{array}{ccc} & & {\left(\rho \beta \right)}_{nf}=\left(1-\phi \right){\left(\rho \beta \right)}_{f}+\phi {\left(\rho \beta \right)}_{p},\;\;\;\;\;\;\;\;\; \end{array}$$(13)$$\begin{array}{ccc} & & \frac{\mu_{nf}}{\mu_{f}}=\frac{1}{1-34.87{\left(\frac{d_{p}}{d_{f}}\right)}^{-0.3}\phi^{1.03}},\;\;\;\;\;\;\;\;\;\; \end{array}$$(14)knfkf=1+4.4ReB0.4Pr0.66TTfr10kpkf0.03ϕ0.66.  
where ReB is defined as
(15)ReB=ρfuBdpμf,uB=2kbTπμfdp2.The molecular diameter of water ($d_{f}$) is given as [26]
(16)$$d_{f}=0.1{\left[\frac{6M}{N\pi \rho_{f}}\right]}^{\frac{1}{3}}.$$Presently we present the employed non-dimensional variables:(17)$$\begin{array}{ccc} & & \left(X,Y\right)=\frac{\left(x,y\right)}{L},\;\;U_{nf,m}=\frac{u_{nf,m}L}{\alpha_{f}},\;\;V_{nf,m}=\frac{v_{nf,m}L}{\alpha_{f}},\;\;\theta_{nf}=\frac{T_{nf}-T_{c}}{T_{h}-T_{c}},\\ & & \theta_{m}=\frac{T_{m}-T_{c}}{T_{h}-T_{c}},\;\;P=\frac{pL^{2}}{\rho_{f}\alpha_{f}^{2}},\;\;k_{eff}=\varepsilon k_{nf}+\left(1-\varepsilon \right)k_{m},\;\;C_{F}=\frac{1.75}{\sqrt{150}}. \end{array}$$The set scheme leads to the following dimensionless governing equations:

In the nanofluid region: (18)$$\begin{array}{ccc} & & \frac{\partial U_{nf}}{\partial X}+\frac{\partial V_{nf}}{\partial Y}=0,\;\;\;\;\;\;\;\;\;\;\;\;\;\;\;\;\;\; \end{array}$$(19)$$\begin{array}{ccc} & & U_{nf}\frac{\partial U_{nf}}{\partial X}+V_{nf}\frac{\partial U_{nf}}{\partial Y}=-\frac{\partial P}{\partial X}+\mathrm{Pr}\frac{\rho_{f}}{\rho_{nf}}\frac{\mu_{nf}}{\mu_{f}}\left(\frac{\partial^{2}U_{nf}}{\partial x^{2}}+\frac{\partial^{2}U_{nf}}{\partial Y^{2}}\right),\\ & & U_{nf}\frac{\partial V_{nf}}{\partial X}+V_{nf}\frac{\partial V_{nf}}{\partial Y}=-\frac{\partial P}{\partial Y}+\mathrm{Pr}\frac{\rho_{f}}{\rho_{nf}}\frac{\mu_{nf}}{\mu_{f}}\left(\frac{\partial^{2}V_{nf}}{\partial X^{2}}+\frac{\partial^{2}V_{nf}}{\partial Y^{2}}\right) \end{array}$$(20)$$\begin{array}{ccc} & & \;\;+\frac{{\left(\rho \beta \right)}_{nf}}{\rho_{nf}\beta_{f}}Ra\;\mathrm{Pr}\;\theta_{nf},\;\;\;\;\;\;\;\;\;\;\; \end{array}$$(21)$$\begin{array}{ccc} & & U_{nf}\frac{\partial \theta_{nf}}{\partial X}+V_{nf}\frac{\partial \theta_{nf}}{\partial Y}=\frac{{\left(\rho C_{p}\right)}_{f}}{{\left(\rho C_{p}\right)}_{nf}}\frac{k_{nf}}{k_{f}}\left(\frac{\partial^{2}\theta_{nf}}{\partial X^{2}}+\frac{\partial^{2}\theta_{nf}}{\partial Y^{2}}\right).\;\;\;\; \end{array}$$

In the porous region:(22)$$\begin{array}{ccc} & & \frac{\partial U_{m}}{\partial X}+\frac{\partial V_{m}}{\partial Y}=0, \end{array}$$
$$\begin{array}{ccc} & & \frac{1}{\varepsilon^{2}}\left(U_{m}\frac{\partial U_{m}}{\partial X}+V_{m}\frac{\partial U_{m}}{\partial Y}\right)=-\frac{\partial P}{\partial X}+\frac{\rho_{f}}{\rho_{nf}}\frac{\mu_{nf}}{\mu_{f}}\frac{\mathrm{Pr}}{\varepsilon }\left(\frac{\partial^{2}U_{m}}{\partial X^{2}}+\frac{\partial^{2}U_{m}}{\partial Y^{2}}\right) \end{array}$$
(23)$$\begin{array}{ccc} & & \;\;-\frac{\rho_{f}}{\rho_{nf}}\frac{\mu_{nf}}{\mu_{f}}\frac{\mathrm{Pr}}{Da}U_{m}-\frac{C_{F}\sqrt{U_{m}^{2}+V_{m}^{2}}}{\sqrt{Da}}\frac{U_{m}}{\varepsilon^{3/2}},\\ & & \frac{1}{\varepsilon^{2}}\left(U_{m}\frac{\partial V_{m}}{\partial X}+V_{m}\frac{\partial V_{m}}{\partial Y}\right)=-\frac{\partial P}{\partial Y}+\frac{\rho_{f}}{\rho_{nf}}\frac{\mu_{nf}}{\mu_{f}}\frac{\mathrm{Pr}}{\varepsilon }\left(\frac{\partial^{2}V_{m}}{\partial X^{2}}+\frac{\partial^{2}V_{m}}{\partial Y^{2}}\right) \end{array}$$
(24)$$\begin{array}{ccc} & & \;\;-\frac{\rho_{f}}{\rho_{nf}}\frac{\mu_{nf}}{\mu_{f}}\frac{\mathrm{Pr}}{Da}V_{m}-\frac{C_{F}\sqrt{U_{m}^{2}+V_{m}^{2}}}{\sqrt{Da}}\frac{V_{m}}{\varepsilon^{3/2}}+\frac{{\left(\rho \beta \right)}_{nf}}{\rho_{nf}\beta_{f}}Ra\;\mathrm{Pr}\;\theta_{m}, \end{array}$$
(25)$$\begin{array}{ccc} & & \frac{1}{\varepsilon }\left(U_{m}\frac{\partial \theta_{m}}{\partial X}+V_{m}\frac{\partial \theta_{m}}{\partial Y}\right)=\frac{k_{eff}}{k_{f}}\frac{{\left(\rho C_{p}\right)}_{f}}{{\left(\rho C_{p}\right)}_{nf}}\left(\frac{\partial^{2}\theta_{m}}{\partial X^{2}}+\frac{\partial^{2}\theta_{m}}{\partial Y^{2}}\right).\;\; \end{array}$$The dimensionless boundary conditions of Equations (Equation 18)–(25) are: (26)$$\begin{array}{ccc} \mathrm{In}\mathrm{the}\mathrm{boundary}\mathrm{of}\mathrm{inner}\mathrm{cylinder}:\;\;U & = & V=0,\;\;\theta_{m}=1,\; \end{array}$$(27)$$\begin{array}{ccc} \mathrm{In}\mathrm{the}\mathrm{boundary}\mathrm{of}\mathrm{outer}\mathrm{cylinder}:\;\;U & = & V=0,\;\;\theta_{nf}=0, \end{array}$$
and the dimensionless boundary forms toward the interface between the nanofluid and porous layers will be obtained from (1) the continuity of tangential and normal velocities, (2) shear and normal stresses, (3) temperature and the heat flux that crossed the central interface and allowing an identical dynamic viscosity ($\mu_{nf}=\mu_{m}$) into both layers. Therefore, the interface dimensionless boundary conditions can be addressed as the following: (28)$$\begin{array}{ccc} \theta_{nf}{|}_{Y=D^{+}} & = & \theta_{m}{|}_{Y=D^{-}},\;\; \end{array}$$(29)$$\begin{array}{ccc} \frac{\partial \theta_{nf}}{\partial Y}{|}_{Y=D^{+}} & = & \frac{k_{eff}}{k_{nf}}\frac{\partial \theta_{m}}{\partial Y}{|}_{Y=D^{-}}, \end{array}$$(30)$$\begin{array}{ccc} U_{nf}{|}_{Y=D^{+}} & = & U_{m}{|}_{Y=D^{-}},\;\; \end{array}$$(31)$$\begin{array}{ccc} V_{nf}{|}_{Y=D^{+}} & = & V_{m}{|}_{Y=D^{-}},\;\; \end{array}$$
here *D* denotes the porous layer’s thickness, and the subscripts + and − indicated that the corresponding measures are estimated while addressing the interface of the nanofluid and porous layers, respectively. $Ra=\frac{g\beta_{f}\left(T_{h}-T_{c}\right)L^{3}}{\nu_{f}\alpha_{f}}$ and $\mathrm{Pr}=\frac{\nu_{f}}{\alpha_{f}}$ signify the Rayleigh number and Prandtl number related to the used base liquid.

Local Nusselt numbers ($Nu_{nf}$ and $Nu_{i}$) at the inner cylinder and interface wall are, respectively, as follows:(32)$$Nu_{nf}=\frac{k_{eff}}{k_{f}}{\left(\frac{\partial \theta }{\partial n}\right)}_{n}.$$
(33)$$Nu_{i}=\frac{k_{nf}}{k_{f}}{\left(\frac{\partial \theta }{\partial D}\right)}_{D},$$
here *n* and *D* determine the entire length of the inner cylinder heat source and interface wall, respectively.

Lastly, the average Nusselt number at the inner heated cylinder is addressed by the following:(34)$$\begin{array}{c} {\bar{Nu}}_{nf}=\int_{0}^{n}Nu_{nf}\;dn, \end{array}$$The entropy production related to the nanofluid layer is provided by [27,28,29]:(35)$$\begin{array}{c} S_{nf}=\frac{k_{nf}}{T_{0}^{2}}\left[{\left(\frac{\partial T}{\partial x}\right)}^{2}+{\left(\frac{\partial T}{\partial y}\right)}^{2}\right]+\frac{\mu_{nf}}{T_{0}}\left[2{\left(\frac{\partial u}{\partial x}\right)}^{2}+2{\left(\frac{\partial v}{\partial y}\right)}^{2}+{\left(\frac{\partial u}{\partial x}+\frac{\partial v}{\partial x}\right)}^{2}\right]. \end{array}$$While the entropy production-related to the porous layer is addressed by [30]:(36)$$\begin{array}{ccc} S_{m} & = & \frac{k_{eff}}{T_{0}^{2}}\left[{\left(\frac{\partial T}{\partial x}\right)}^{2}+{\left(\frac{\partial T}{\partial y}\right)}^{2}\right]+\frac{\mu_{f}}{T_{0}}\left[2{\left(\frac{\partial u}{\partial x}\right)}^{2}+2{\left(\frac{\partial v}{\partial y}\right)}^{2}+{\left(\frac{\partial u}{\partial x}+\frac{\partial v}{\partial x}\right)}^{2}\right]\\ & & +\;\frac{\mu_{f}}{KT_{0}}\left(u^{2}+v^{2}\right). \end{array}$$In the dimensionless pattern, Equation (Equation 35) can be signified as:(37)$$\begin{array}{ccc} & & S_{\mathrm{GEN},nf}=\frac{k_{nf}}{k_{f}}\left[{\left(\frac{\partial \theta }{\partial X}\right)}^{2}+{\left(\frac{\partial \theta }{\partial Y}\right)}^{2}\right]\\ & & \;\;+\frac{\mu_{nf}}{\mu_{f}}N_{nf}\left\{2\left[{\left(\frac{\partial U}{\partial X}\right)}^{2}+{\left(\frac{\partial V}{\partial Y}\right)}^{2}\right]+{\left(\frac{\partial^{2}U}{\partial Y^{2}}+\frac{\partial^{2}V}{\partial X^{2}}\right)}^{2}\right\}, \end{array}$$
where $N_{nf}$ denotes the irreversibility distribution ratio within the nanofluid region and can be represented by:(38)$$\begin{array}{c} N_{nf}=\frac{\mu_{f}T_{0}}{k_{f}}{\left(\frac{\alpha_{f}}{L(\Delta T)}\right)}^{2}, \end{array}$$
and $S_{\mathrm{GEN},nf}$ explains the dimensionless entropy production rate:(39)$$\begin{array}{c} S_{\mathrm{GEN},nf}=S_{\mathrm{gen},nf}\frac{T_{0}^{2}L^{2}}{k_{f}{\left(\Delta T\right)}^{2}}. \end{array}$$The terms of Equation (Equation 39) will be divided based on the following scheme:(40)$$\begin{array}{c} S_{\mathrm{GEN},nf}=S_{nf,\theta }+S_{nf,\Psi }, \end{array}$$
here, $S_{nf,\theta }$ and $S_{nf,\Psi }$ obtain the entropy generation regarding the heat transfer irreversibility (HTI) and fluid friction irreversibility (FFI) of the nanofluid layer, respectively.
(41)$$\begin{array}{ccc} & & S_{nf,\theta }=\frac{k_{nf}}{k_{f}}\left[{\left(\frac{\partial \theta }{\partial X}\right)}^{2}+{\left(\frac{\partial \theta }{\partial Y}\right)}^{2}\right],\;\;\;\;\;\;\;\;\;\;\;\; \end{array}$$
(42)$$\begin{array}{ccc} & & S_{nf,\Psi }=\frac{\mu_{nf}}{\mu_{f}}N_{nf}\left\{2\left[{\left(\frac{\partial U}{\partial X}\right)}^{2}+{\left(\frac{\partial V}{\partial Y}\right)}^{2}\right]+{\left(\frac{\partial^{2}U}{\partial Y^{2}}+\frac{\partial^{2}V}{\partial X^{2}}\right)}^{2}\right\}.\;\; \end{array}$$Into the dimensionless structure, Equation (Equation 36) remains exposed as:(43)$$\begin{array}{ccc} & & S_{\mathrm{GEN},m}=\frac{k_{eff}}{k_{f}}\left[{\left(\frac{\partial \theta }{\partial X}\right)}^{2}+{\left(\frac{\partial \theta }{\partial Y}\right)}^{2}\right]+\frac{\mu_{nf}}{\mu_{f}}N_{m}\times \\ & & \;\left\{Da\left[2\left({\left(\frac{\partial U}{\partial X}\right)}^{2}+{\left(\frac{\partial V}{\partial Y}\right)}^{2}\right)+{\left(\frac{\partial^{2}U}{\partial Y^{2}}+\frac{\partial^{2}V}{\partial X^{2}}\right)}^{2}\right]+\left(U^{2}+V^{2}\right)\right\}, \end{array}$$
where $N_{m}=\frac{\mu_{f}T_{0}}{k_{f}}\left(\frac{\alpha_{f}^{2}}{K{\left(\Delta T\right)}^{2}}\right)$ is the irreversibility distribution ratio in the porous layer and $S_{\mathrm{GEN},m}=S_{\mathrm{gen},m}\frac{T_{0}^{2}L^{2}}{k_{f}{\left(\Delta T\right)}^{2}}$.

The terms of Equation (Equation 43) can be separated into the following form:(44)$$\begin{array}{c} S_{\mathrm{GEN},m}=S_{m,\theta }+S_{m,\Psi }, \end{array}$$
where $S_{m,\theta }$ and $S_{m,\Psi }$ are the entropy generation due to heat transfer irreversibility (HTI) and fluid friction irreversibility (FFI) of the porous layer, respectively.
(45)$$\begin{array}{ccc} & & S_{m,\theta }=\frac{k_{eff}}{k_{f}}\left[{\left(\frac{\partial \theta }{\partial X}\right)}^{2}+{\left(\frac{\partial \theta }{\partial Y}\right)}^{2}\right],\;\;\;\;\;\;\;\;\;\;\;\;\;\\ & & S_{m,\Psi }=\frac{\mu_{nf}}{\mu_{f}}N_{m}\times \;\;\;\;\;\;\;\;\;\;\;\;\; \end{array}$$
(46)$$\begin{array}{ccc} & & \left\{Da\left[2\left({\left(\frac{\partial U}{\partial X}\right)}^{2}+{\left(\frac{\partial V}{\partial Y}\right)}^{2}\right)+{\left(\frac{\partial^{2}U}{\partial Y^{2}}+\frac{\partial^{2}V}{\partial X^{2}}\right)}^{2}\right]+\left(U^{2}+V^{2}\right)\right\}. \end{array}$$In the dimensionless form, the local entropy generation can be expressed as:(47)$$\begin{array}{c} S_{\mathrm{GEN}}=S_{\mathrm{GEN},nf}+S_{\mathrm{GEN},m}. \end{array}$$The global entropy generation (GEG) is obtained by integrating Equation (Equation 47) over the domain
(48)$$\begin{array}{c} \mathrm{GEG}=\int S_{\mathrm{GEN}}\;dXdY=\int S_{\mathrm{GEN},nf}\;dXdY+\int S_{\mathrm{GEN},m}\;dXdY. \end{array}$$The Bejan number Be is defined as:(49)$$\begin{array}{c} Be=\frac{\int S_{nf,\theta }\;dXdY+\int S_{m,\theta }\;dXdY}{\mathrm{GEG}}. \end{array}$$When Be>0.5, the HTI is dominant, while when Be<0.5, the FFI is dominant.

## 3. Numerical Method and Validation

The governing dimensionless equations Equations (Equation 18)–(25) ruled with the boundary conditions Equations (Equation 26)–(31) are solved by the Galerkin weighted residual finite element technique. The computational region does discretize toward small triangular portions, as shown in Figure 2.

These small triangular Lagrange components with various forms are applied to each flow variable within the computational region. Residuals for each conservation equation exist and are accomplished through substituting the approximations within the governing equations. The Newton–Raphson iteration algorithm is adopted for clarifying the nonlinear expressions into the momentum equations. The convergence from the current numerical solution is considered, while the corresponding error toward each of the variables fills the resulting convergence criteria:$$\begin{array}{c} \left|\frac{\Gamma^{i+1}-\Gamma^{i}}{\Gamma^{i+1}}\right|\leq 10^{-6}, \end{array}$$

For confirming the independence regarding the existing numerical solution at the grid size of the numerical region, different grid sizes are used to determine the average Nusselt number (${\bar{Nu}}_{nf}$), average temperature ($\theta_{\mathrm{avg}}$) and Bejan number (Be) for the case of $Ra=10^{6}$, $Da=10^{-3}$, ϕ=0.02, D=0.5 and ε=0.5. The outcomes given in Table 1 designate irrelevant variations for the G5 grids and beyond. Hence, for all calculations into this work and similar obstacles to this subsection, the G5 uniform grid is applied.

Regarding the validation for the numerical data, the current results have been compared with earlier experimental and numerical outcomes achieved by Kuehn and Goldstein [31], as shown in Figure 3 concerning the natural convection heat transfer case within a horizontal concentric annulus saturated with pure fluid (water). Furthermore, the results are considered among earlier declared experimental results obtained by Beckermann et al. [32] via a natural convection problem inside a square cavity having both fluid and porous layers, as implemented in Figure 4. As explained by the earlier performed comparisons, the numerical outcomes of the present numerical code are essential to a high degree of reliability.

## 4. Results and Discussion

The outcomes described by streamlines, isotherms, and isentropic distributions are addressed within this segment. We have modified the following four parameters; Darcy number ($10^{-6}\leq Da\leq 10^{-2}$), nanoparticle volume fraction (0≤ϕ≤0.04), dimensionless porous layer length (0.4≤D≤0.7) and the porosity of the medium (0.2≤ε≤0.8). The values of the Rayleigh number and Prandtl number are fixed at $Ra=10^{6}$ and Pr=4.623, respectively. Table 2 displays the thermophysical properties of the base fluid (water) and the solid Al${}_{2}$O${}_{3}$ phases at T=310 K.

The reliance of the streamlines, isotherms, and isentropic lines on the Darcy numbers (Da) when ϕ=0.02, D=0.5 and ε=0.5 has been shown in Figure 5. It is remarked that as Da increases from $10^{-5}$ to $10^{-2}$, the absolute value of the streamline’s maximum ${\left|\psi \right|}_{max}$ is strongly increasing. The reason is the altitude porous resistance of the nanofluid flow at lower Da. Thus, the streamlines’ intensity around a circular cylinder within an annulus raises as Da increases. The isothermal lines filled the annulus at the lower Darcy parameter $Da=10^{-5}$ compared to the higher Darcy parameter $Da=10^{-2}$. The isotherms’ strength is rising across the top area of an annulus according to the influence in Da. Thus, the intensity of the isentropic lines is strongly rising as Da increases. At the lower Darcy parameter, $Da\leq 10^{-4}$, isentropic lines are distributed around the heater of an inner circular cylinder with fewer lines near the cold outer wall. Furthermore, an increment in Da augments the isentropic lines around the top area of an annulus, and it seems that the distributions of the isentropic lines are following the strength of the isothermal and streamlines contours.

Figure 6 presents the influences of Da toward the local velocity, local temperature, and local interface Nusselt number on the interface wall within nanofluid-porous layers, *W* at ϕ=0.02 and ε=0.5. In Figure 6a, it is observed that there are three peaks on the local velocity *V* across *W* around the locations of W=0.5,1, and 1.5, in which the local velocity is (V≤0). Further, the local velocity, *V* is improving clearly as Da boosts. Figure 6b shows fluctuations in the local temperature across *W* below the variations on Da. When (0.5≤W≤1), the value of $Da=10^{-4}$ provides the highest values of the local temperature and in the other parts of *W*, the highest values of the local temperature are obtained at $Da=10^{-5}$ and the lowest at $Da\geq 10^{-3}$. The results are returning to the distributions of the isothermal lines through an annulus, as reported above in Figure 5. In Figure 6c, the local interface Nusselt number across the interface wall *W* showed fluctuations under the effects of Da.

Figure 7 depicts the impacts of porosity parameter ε, at the streamlines, isotherms, and isentropic lines at $Da=10^{-3}$, ϕ=0.02 and D=0.5. It is observed that the value of ${\left|\psi \right|}_{max}$ increases by 32.37%, as ε raises from 0.2 to 0.8. Thus, the streamline contours are affected by the variations on ε. There are minor reductions in the isothermal lines according to an increase in the value of ε. Whilst, the contours of the isentropic lines are clearly improved within an annulus according to an increase in the value of ε. Figure 8 presents the influences of ε on the local velocity, local temperature, and local interface Nusselt number on the interface surface within nanofluid-porous layers, *W* at $Da=10^{-3}$, ϕ=0.02 and D=0.5. The local velocity, *V*, showed a clear increase from W=0.5 to 1, as ε and in the other parts of the interface wall, there is almost no change on the values of *V* according to the variations on ε. Further, there is a slight enhancement in the local temperature and local interface Nusselt number according to a reduction in ε. The physical representations of these results are returning to the medium’s porosity definition as it is characterized from the ratio of pore volume and the total volume of material, and an increase in ε towards the unity represents the free fluid.

The dependency of the streamlines, isotherms, and isentropic lines on the various dimensionless length of porous layer *D* at $Da=10^{-3}$, ϕ=0.02 and ε=0.5 have been shown in Figure 9. The first remark is that ${\left|\psi \right|}_{max}$ reduces by 25% as *D* raises from 0.4 to 0.7. The reason returns to the nanofluid flow’s porous resistance within an annulus at an extra length of a porous layer *D*. There is a slight enhancement in the isothermal contours according to an increase in *D*. Thus, the contours of isentropic lines are influenced by the variations on the length of a porous layer *D*. Figure 10 presents the local velocity, local temperature, and local interface Nusselt number at the interface wall between nanofluid-porous layers under the variations on *D* at $Da=10^{-3}$, ϕ=0.02 and ε=0.5. It is observed that the lower value of *D* gives the highest values of the local velocity at the interface wall. Due to the expansion of *D*, the values of local velocity fluctuate between positive and negative values, as shown previously for an interface wall’s location over the streamlines (Figure 9). Further, the local temperature’s highest peak across the interface wall appears at the lowest value of D=0.4. The profiles of ${\bar{Nu}}_{i}$ are decreasing according to an increment in *D*.

Figure 11 introduces the changes of average Nusselt number (${\bar{Nu}}_{nf}$), average temperature ($\theta_{\mathrm{avg}}$), and Bejan number (Be) during various Da and ϕ at ε=0.5 and D=0.5. It is remarked that ${\bar{Nu}}_{nf}$ is slightly enhanced as ϕ increases and ${\bar{Nu}}_{nf}$ is strongly improved as Da powers. There are minor changes on the $\theta_{\mathrm{avg}}$ under the variations on ϕ. A reduction in Da augments the value of $\theta_{\mathrm{avg}}$. Further, the value of Be showed a similar tendency as $\theta_{\mathrm{avg}}$ under the influences of ϕ and Da. The values of Be increase as Da decreases, and there is a minor enhancement in Be under the augmentation in ϕ. The outcomes show improved heat transfer through high porous resistance (lower Da).

Figure 12 indicates the dependency of ${\bar{Nu}}_{nf}$, $\theta_{\mathrm{avg}}$, and Bejan number Be on Da and ε at ϕ=0.02 and D=0.5. In this figure, an augmentation in Da raises ${\bar{Nu}}_{nf}$ and declines the values of $\theta_{\mathrm{avg}}$ and Be. The physical reason returns to the high porous resistance of the nanofluid movements at a lower Darcy parameter. A decrease in Da enhances the heat transfer within a circular cylinder. Further, there are minor changes in ${\bar{Nu}}_{nf}$, $\theta_{\mathrm{avg}}$ and Be under the variations of ε. It is determined that the impacts of Da on ${\bar{Nu}}_{nf}$, $\theta_{\mathrm{avg}}$ and Be are high enough, in which the impacts of ϕ and ε can be neglected.

Figure 13 shows the influences of ϕ and ε on ${\bar{Nu}}_{nf}$, $\theta_{\mathrm{avg}}$ and Be at $Da=10^{-3}$ and D=0.5. Here, an increment in ϕ improves the values of ${\bar{Nu}}_{nf}$ and declines the values of $\theta_{\mathrm{avg}}$ and Be. Furthermore, there are almost no changes in ${\bar{Nu}}_{nf}$, $\theta_{\mathrm{avg}}$ and Be below the variations of ε. The physical reason returns to low porous resistance according to the alterations on ε. In Figure 14, for any value of *D*, an increment in Da is still enhancing ${\bar{Nu}}_{nf}$ and reduces $\theta_{\mathrm{avg}}$ and Be. Further, the values of ${\bar{Nu}}_{nf}$, $\theta_{\mathrm{avg}}$ and Be have little changes according to the variation on the dimensionless length of porous layer *D* at different values of Da.

## 5. Conclusions

This study applied the entropy generation on nanofluid superposed horizontal annulus porous layers via the fluid flow and heat transfer characteristics. Here, the values of local velocity, local temperatures, local Nusselt numbers, average Nusselt number, average temperature, and Bejan number are measured in the interface surface within nanofluid-porous layers. The following points are summarizing the main findings:Increasing the Darcy parameter from $10^{-5}$ to $10^{-2}$ powers the value of streamlines’ maximum and strengthens the intensity of the isotherms and isentropic lines within an annulus due to higher porous resistance at a lower Darcy parameter.An augmentation in the Darcy parameter boosts the local velocity and average Nusselt number and reduces the average temperature and Bejan number.Increasing the porosity parameter augments the streamlines’ maximum by 32.37% and improves the contours of isentropic lines within an annulus.The value of streamlines’ maximum reduces by 25% as the length of a porous layer raises from 0.4 to 0.7.The highest values of the local velocity and local temperature correspond with lower lengths of a porous layer.The profiles of the local Nusselt number decrease according to an increase in the length of a porous layer.Increasing the solid volume fraction improves the values of the average Nusselt number and reduces the average temperature and Bejan number.

## Figures and Tables

**Figure 1 entropy-23-01237-f001:**
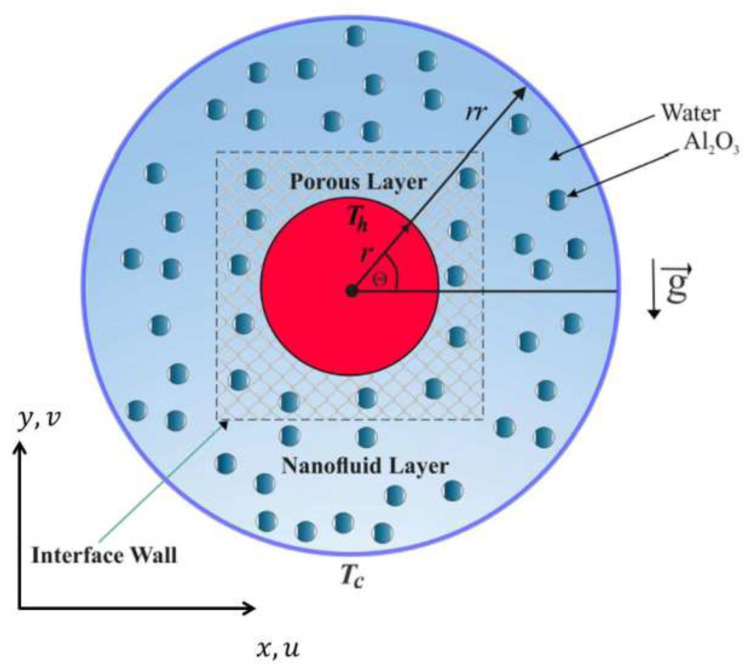
Schematic representation concerning the convection flow in annulus composite.

**Figure 2 entropy-23-01237-f002:**
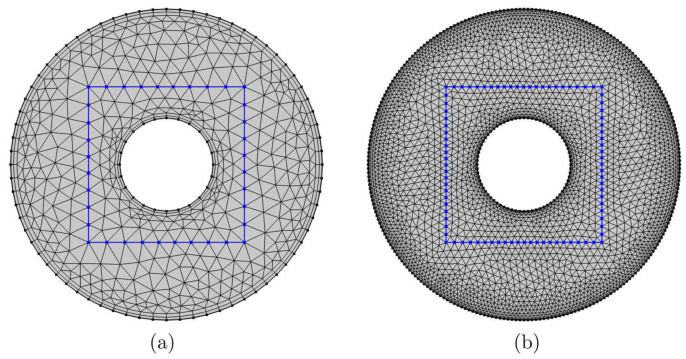
Framework configuration of the FEM for the grid dimension of (**a**) 958 and (**b**) 6026 components.

**Figure 3 entropy-23-01237-f003:**
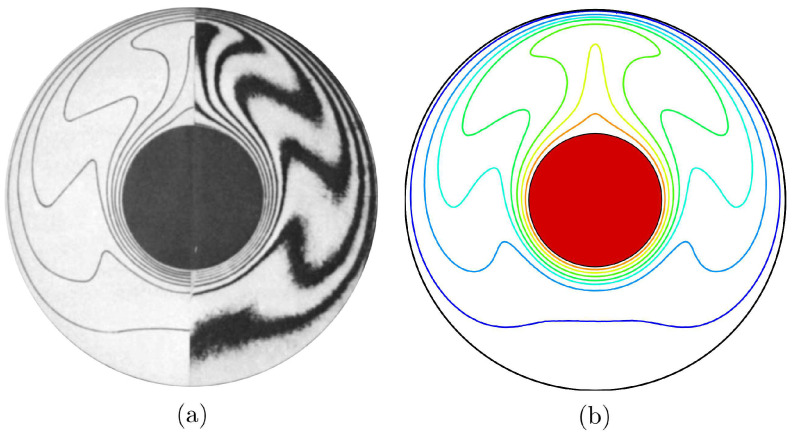
Isotherms of (**a**) Kuehn and Goldstein [31]; numerical result (left) and experimental result (right), and (**b**) present study for $Ra=5\times 10^{4}$ and Pr=0.7.

**Figure 4 entropy-23-01237-f004:**
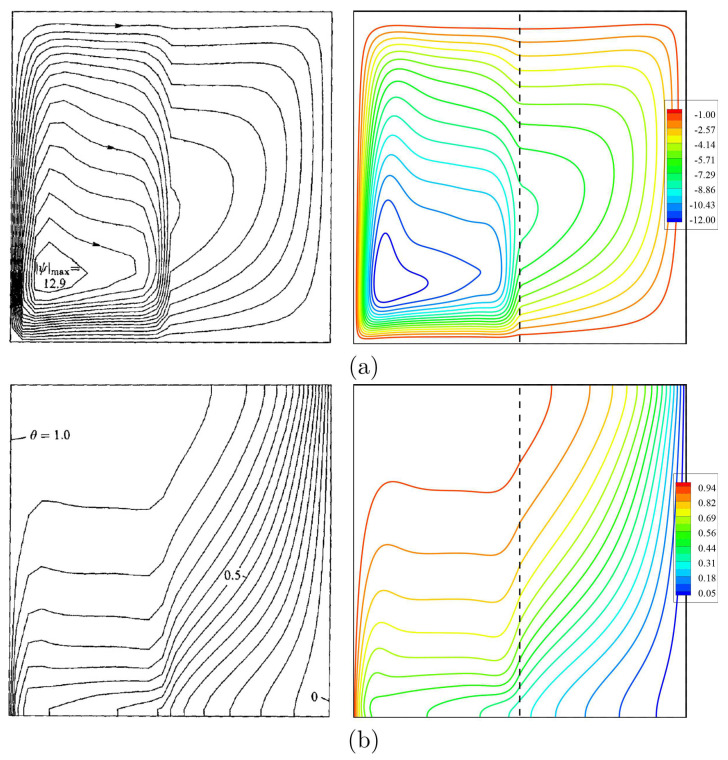
(**a**) Streamlines of Beckermann et al. [32] (left) and the wpresent study (right); (**b**) isotherms for $Ra=3.70\times 10^{6}$, $Da=1.370\times 10^{-5}$, ε=0.9, D=0.5, N=0, $\frac{k_{eff}}{k_{f}}=1.362$ and Pr=6.44.

**Figure 5 entropy-23-01237-f005:**
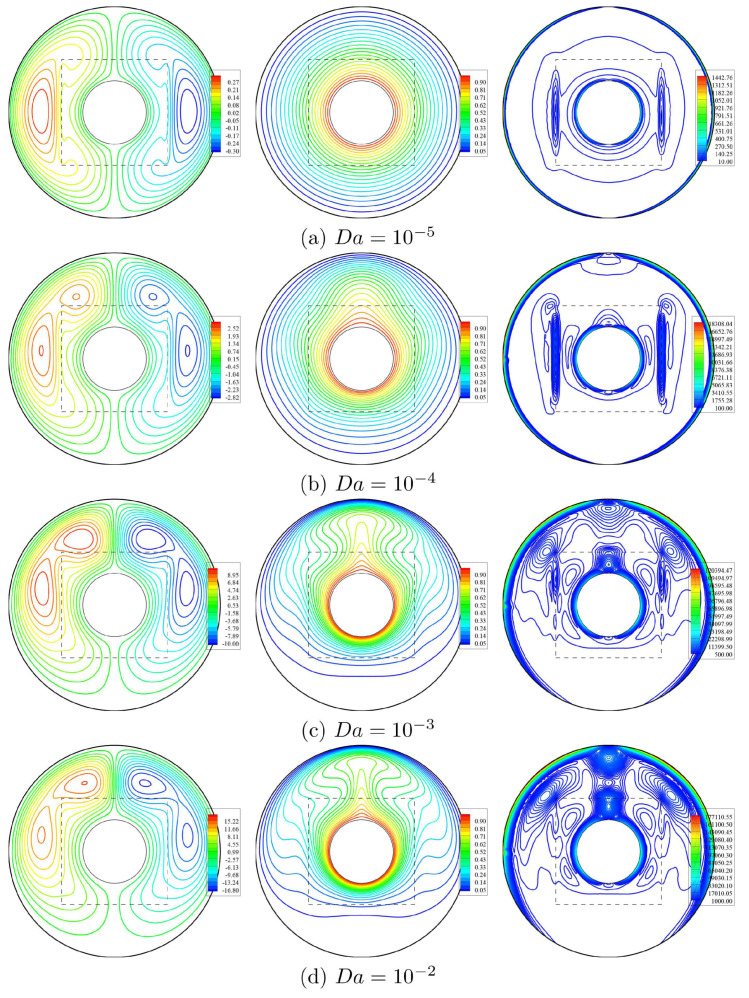
Streamlines (**left**), isotherms (**middle**), and isentropic lines (**right**) with various Darcy numbers (Da); ϕ=0.02, ε=0.5 and D=0.5.

**Figure 6 entropy-23-01237-f006:**
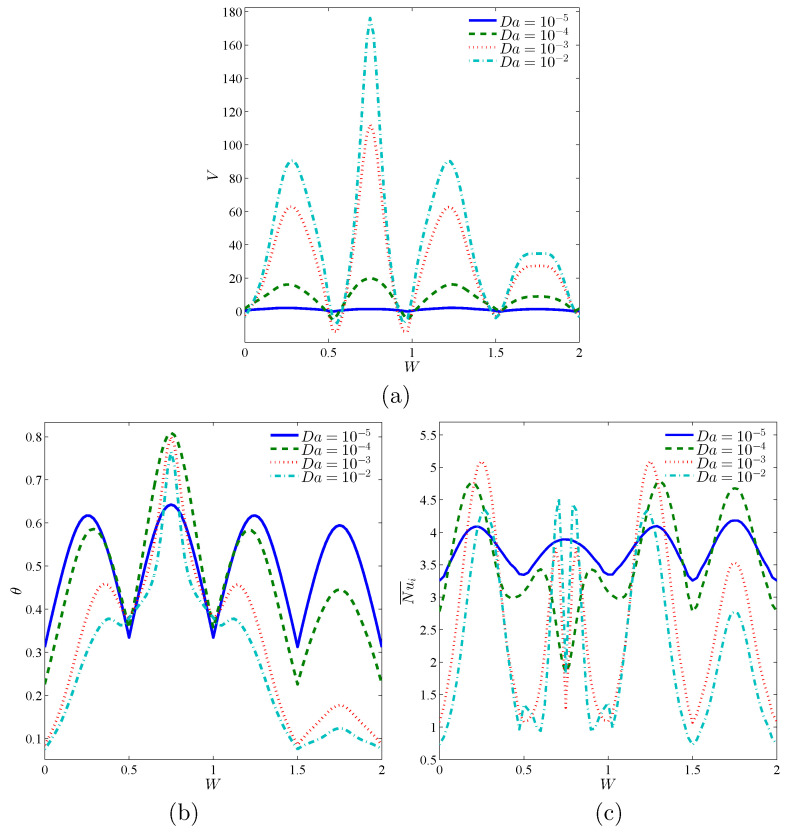
Local velocity (**a**), local temperature (**b**) and local interface Nusselt number (**c**) at the interface wall between nanofluid-porous layers for different Da; ϕ=0.02, ε=0.5 and D=0.5.

**Figure 7 entropy-23-01237-f007:**
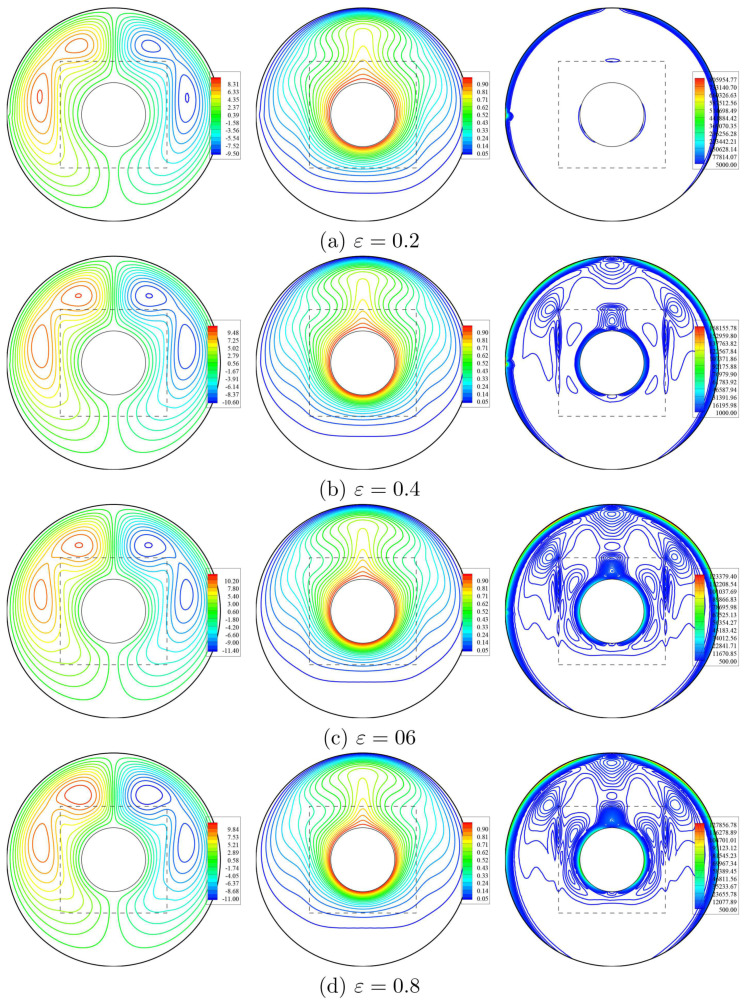
Streamlines (**left**), isotherms (**middle**), and isentropic lines (**right**) with various porosity of the medium (ε); $Da=10^{-3}$, ϕ=0.02 and D=0.5.

**Figure 8 entropy-23-01237-f008:**
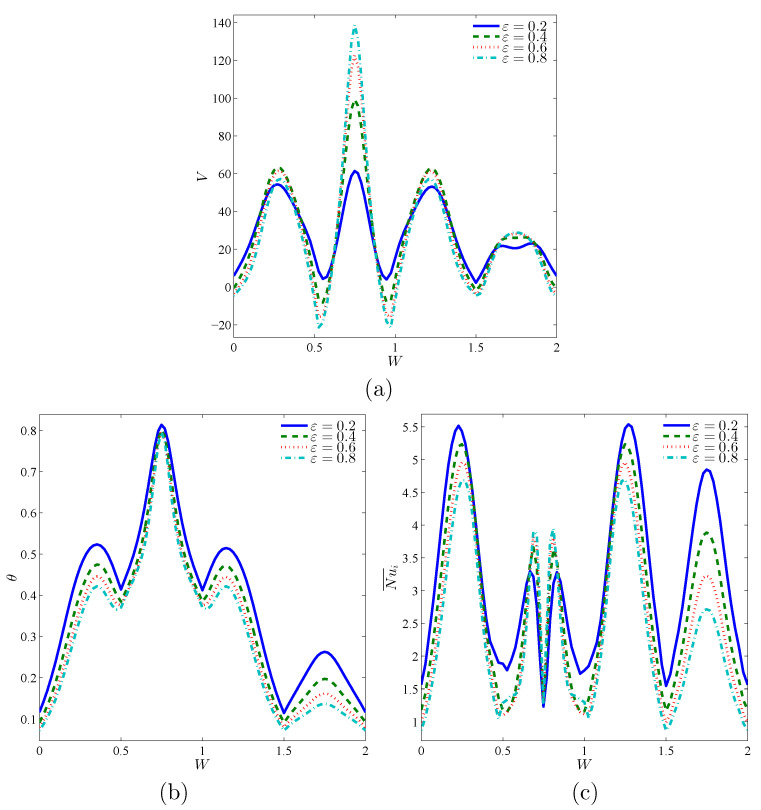
Local velocity (**a**), local temperature (**b**) and local interface Nusselt number (**c**) at the interface wall between nanofluid-porous layers for different ε; $Da=10^{-3}$, ϕ=0.02 and D=0.5.

**Figure 9 entropy-23-01237-f009:**
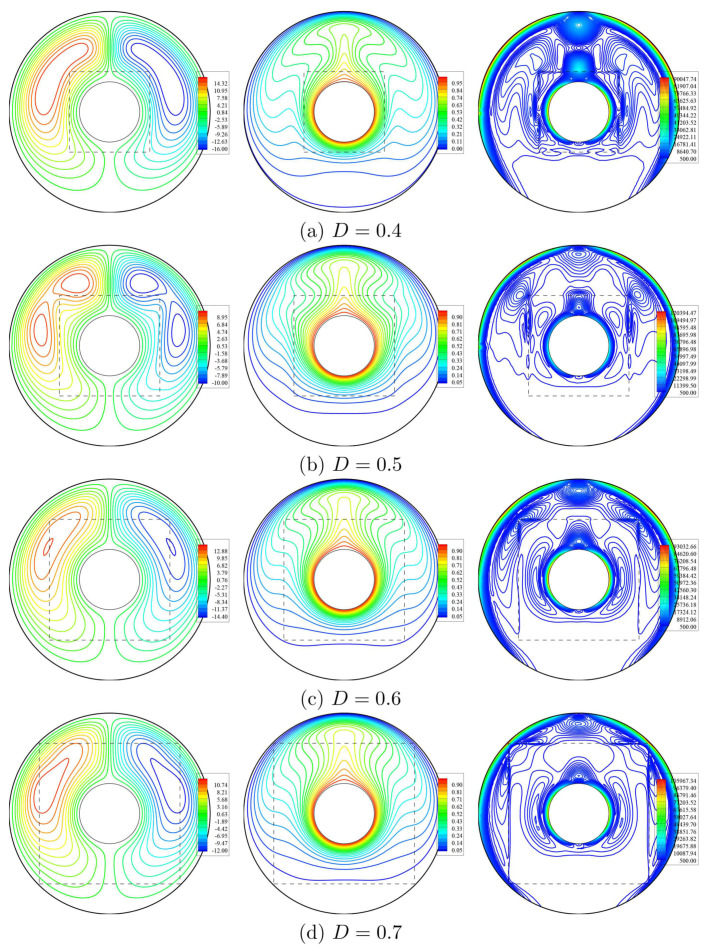
Streamlines (**left**), isotherms (**middle**), and isentropic lines (**right**) with various dimensionless length of porous layer (*D*); $Da=10^{-3}$, ϕ=0.02 and ε=0.5.

**Figure 10 entropy-23-01237-f010:**
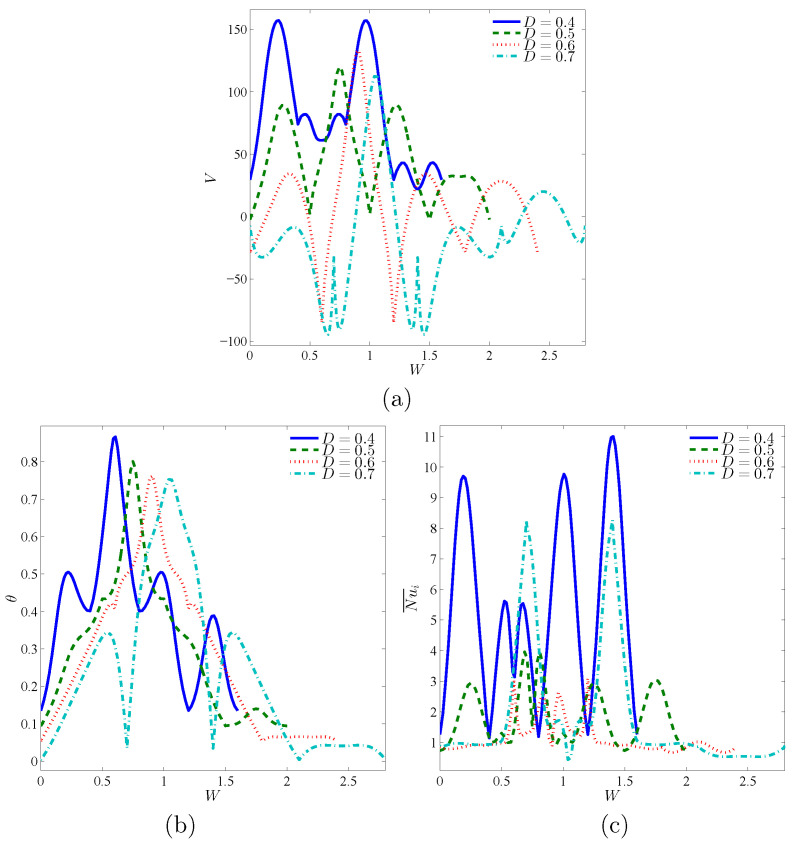
Local velocity (**a**), local temperature (**b**) and local interface Nusselt number (**c**) at the interface wall between nanofluid-porous layers for different *D*; $Da=10^{-3}$, ϕ=0.02 and ε=0.5.

**Figure 11 entropy-23-01237-f011:**
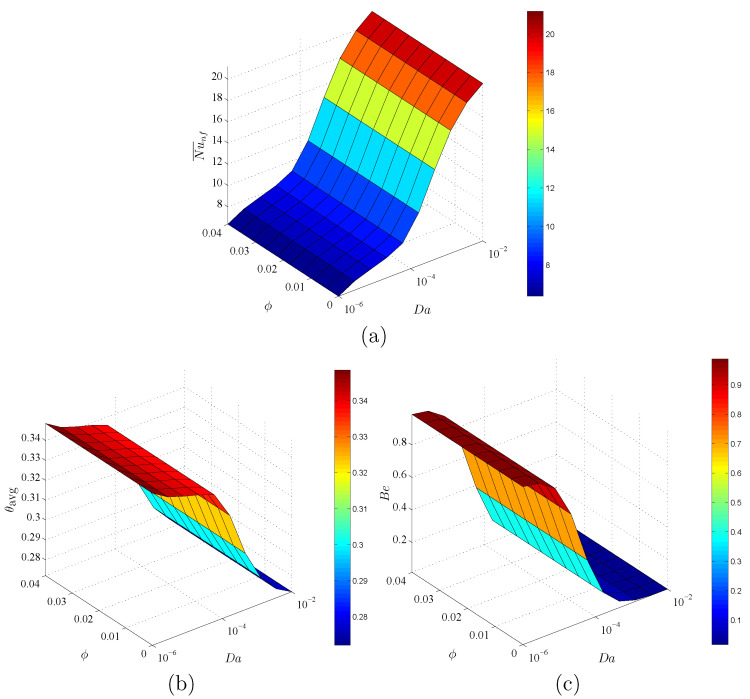
Variations of (**a**) average Nusselt number, (**b**) average temperature and (**c**) Bejan number for different Da and ϕ at ε=0.5 and D=0.5.

**Figure 12 entropy-23-01237-f012:**
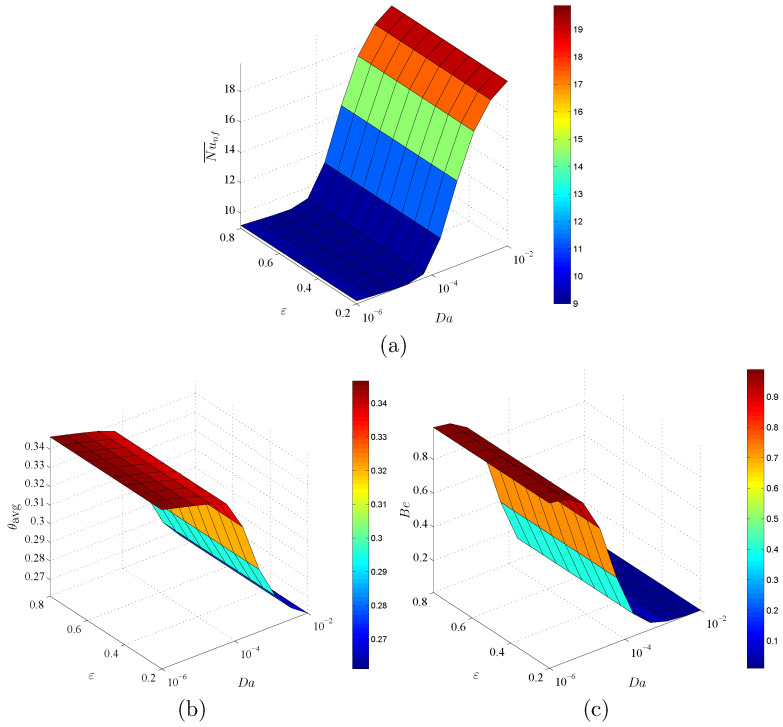
Variations of (**a**) average Nusselt number, (**b**) average temperature and (**c**) Bejan number for different Da and ε at ϕ=0.02 and D=0.5.

**Figure 13 entropy-23-01237-f013:**
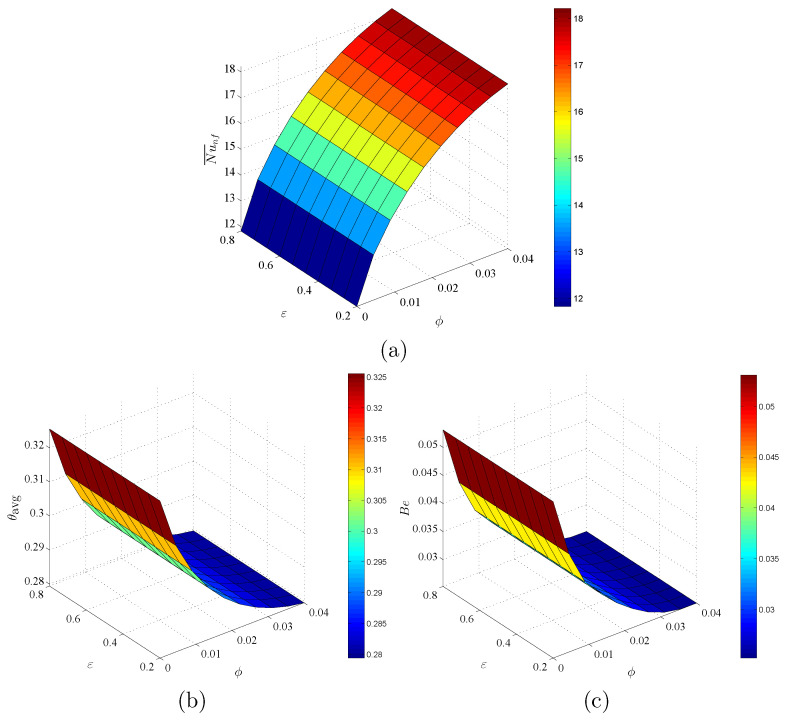
Variations of (**a**) average Nusselt number, (**b**) average temperature and (**c**) Bejan number for different ϕ and ε at $Da=10^{-3}$ and D=0.5.

**Figure 14 entropy-23-01237-f014:**
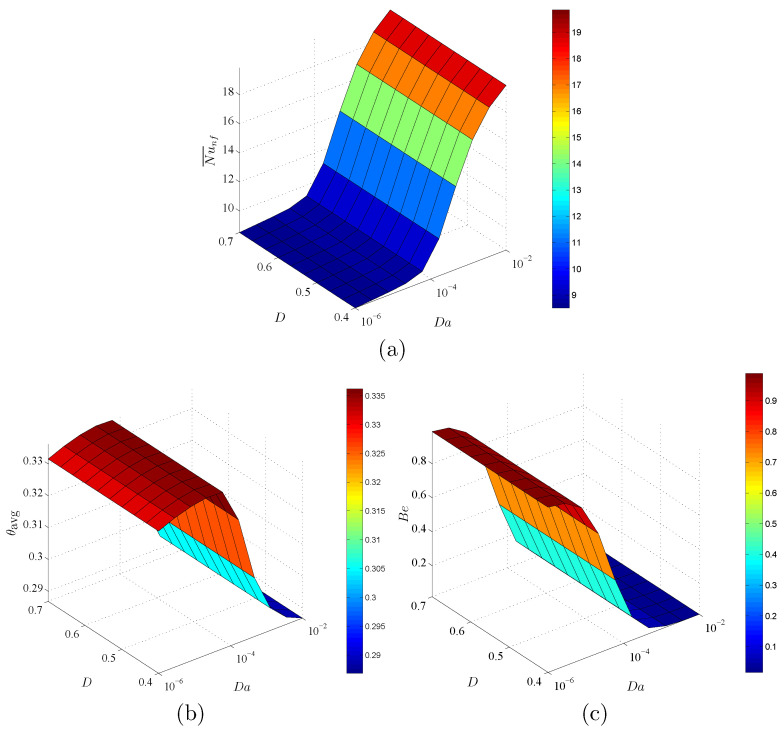
Variations of (**a**) average Nusselt number, (**b**) average temperature and (**c**) Bejan number for different Da and *D* at ϕ=0.02 and ε=0.5.

**Table 1 entropy-23-01237-t001:** Grid testing for ${\bar{Nu}}_{nf}$, $\theta_{\mathrm{avg}}$ and Be at different grid sizes for $Ra=10^{6}$, $Da=10^{-3}$, ϕ=0.02, ε=0.5 and D=0.5.

Grid Size	Number of Elements	${\bar{\mathrm{Nu}}}_{\mathrm{nf}}$	$\theta_{\mathrm{avg}}$	Be
G1	486	19.519	0.30926	0.011058
G2	958	19.525	0.29652	0.012067
G3	1310	19.533	0.29534	0.016238
G4	2008	19.539	0.29143	0.01726
G5	6026	19.539	0.28799	0.018096
G6	17324	19.543	0.28742	0.018244

**Table 2 entropy-23-01237-t002:** Thermophysical characteristics concerning pure liquid (water) and Al${}_{2}$O${}_{3}$ nanoparticles at T=310 K [33].

Physical Properties	Fluid Phase (Water)	Al${}_{2}$O${}_{3}$
$C_{p}\;\left(J/\mathrm{kg}\;K\right)$	4178	765
$\rho \;(\mathrm{kg}/m^{3})$	993	3970
$k\;(W\;m^{-1}\;K^{-1})$	0.628	40
$\beta \times 10^{5}\;\left(1/K\right)$	36.2	0.85
$\mu \times 10^{6}\;\left(\mathrm{kg}/m\;s\right)$	695	–
$d_{p}\;\left(\mathrm{nm}\right)$	0.385	33

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
