# Peer review of "Energy and Entropy Production of Nanofluid within an Annulus Partly Saturated by a Porous Region"

_entropy, 2021, doi:10.3390/e23101237_

Round 1

Reviewer 1 Report

The paper Energy and entropy production of nanofluid within an annulus partly saturated by a porous region is in the scope of Entropy. Before publication the authors should response :

1- What is the engineering benefit of using such reactangular porous geometry in a pipe?

2-  The use of term : "local thermal non-equilibrium model" is not seems proper. There is no such phenomena in calculation of the porous but in a homogenous nanofluid it used which is rare and not clear why.

3-  Add more literature from MDI.

4- trend of figure 11-14 is expected for various Darcey number. What is new in the papers findings ?

Author Response

We thank the respected reviewers for their constructive comments which clearly enhanced the quality of the manuscript. Our replies to the comments are given in the attachment.

Reviewer 2 Report

  • Delete the first three sentences of Introduction Section since they are unnecessary.
  • The first sentence of the Mathematical Formulation should be checked and rewritten.
  • Cite References for Eq. (9).
  • There is no explanation in the text for Fig. 2.
  • Mesh independence study should be added.
  • On Page 12, "Da additions of 10-5 to 10-2" should be checked and rewritten.
  • On page 12, "the absolute value of the streamlines's maximum
    increases strongly by 5500%"
  • On page 12, Explain what a 5500% increase physically means.
  • In Line 6 on Page 12,  "thermos-physical properties" should be "thermophysical properties"
  •  It is necessary to report whether the Prandtl number can be assumed to be a constant (4.623) even though the nanoparticles volume fractions change.  
  • Nomenclature section should be included.
  • Do not use symbols in conclusion section.

Author Response

We thank the respected reviewers for their constructive comments which clearly enhanced the quality of the manuscript. Our revised manuscript has given in the attachment.

Round 2

Reviewer 1 Report

For the enginnering application please give a figure or reference that have same geometry or material configurations.

Author Response

First of all, we thank the respected reviewers for their constructive comments which clearly enhanced the quality of the manuscript. Our replies to the comments are given below:

Referee  

Comment: For the engineering application please give a figure or reference that have same geometry or material configurations.

Reply: Many thanks for the respected reviewer for the above mentioned comment. The engineering application of such a composed cavity can be located in a wide range of industrial and environmental applications like fibrous thermal insulation, solidification, fuel cell, cooling of nuclear fuel debris, solar collectors, and underground storage of radioactive waste are some of the many applications. Also such a study can be useful in the design of the heating system [1].

Fig.1 (In the attached file). Comparison of the local Nusselt number at the outer cylinder and at the interface of the porous medium of [1].

[1] Kumari, M., & Nath, G. (2008). Unsteady natural convection from a horizontal annulus filled with a porous medium. International journal of heat and mass transfer, 51(19-20), 5001-5007.
